# Predictors and outcomes of peritoneal dialysis-related infections due to filamentous molds (MycoPDICS)

Talerngsak Kanjanabuch[1,2,3,4¶]*, Tanawin Nopsopon[5], Tanittha Chatsuwan[6,7], Sirirat Purisinsith[8], David W Johnson[9,10,11], Nibondh Udomsantisuk[6], Guttiga Halue[12¶], Pichet Lorvinitnun[13¶], Pongpratch Puapatanakul[1,3¶], Krit Pongpirul[5], Ussanee Poonvivatchaikarn[14], Sajja Tatiyanupanwong[15¶], Saowalak Chowpontong[16], Rutchanee Chieochanthanakij[17], Oranan Thamvichitkul[18], Worapot Treamtrakanpon[19¶], Wadsamon Saikong[20], Uraiwan Parinyasiri[21¶], Piyatida Chuengsaman[22¶], Phongsak Dandecha[23¶], Jeffrey Perl[24], Kriang Tungsanga[1], Somchai Eiam-Ong[1], Suchai Sritippayawan[25¶], Surasak Kantachuvesiri[26], on behalf of The Advisory Board of Peritoneal Dialysis, Nephrology Society of Thailand[¶]

1 Division of Nephrology, Department of Medicine, Bangkok, Thailand, 2 Center of Excellence in Kidney Metabolic Disorders, Bangkok, Thailand, 3 Peritoneal Dialysis Excellent Center, King Chulalongkorn Memorial Hospital, Bangkok, Thailand, 4 Dialysis Policy and Practice Program (Di3P), Bangkok, Thailand, 5 Department of Preventive and Social Medicine, Bangkok, Thailand, 6 Department of Microbiology, Bangkok, Thailand, 7 Faculty of Medicine, Antimicrobial Resistance and Stewardship Research Unit, Chulalongkorn University, Bangkok, Thailand, 8 Health Department, Bangkok Metropolitan Administration, Bangkok, Thailand, 9 Department of Nephrology, Princess Alexandra Hospital, Brisbane, QLD, Australia, 10 Australasian Kidney Trials Network, University of Queensland, Brisbane, Australia, 11 Translational Research Institute, Brisbane, Australia, 12 Department of Medicine, Phayao Hospital, Phayao, Thailand, 13 Department of Medicine, Sunpasitthiprasong Hospital, Ubon Ratchathani, Thailand, 14 Nephrology Clinic, Nakhon Pathom Hospital, Nakhon Pathom, Thailand, 15 Nephrology Division, Department of Internal Medicine, Chaiyaphum Hospital, Chaiyaphum, Thailand, 16 Division of Nephrology, Department of Medicine, Phra Nakhon Si Ayutthaya Hospital, Phra Nakhon Si Ayutthaya, Thailand, 17 Dialysis Unit, Sawanpracharak Hospital, Nakhon Sawan, Thailand, 18 Dialysis Unit, Sisaket Hospital, Sisaket, Thailand, 19 Department of Medicine, Chaophraya Abhaibhubejhr Hospital, Prachin Buri, Thailand, 20 Continuous Ambulatory Peritoneal Dialysis Clinic, Mukdahan Hospital, Mukdahan, Thailand, 21 Kidney diseases clinic, Department of internal medicines, Songkhla Hospital, Songkhla, Thailand, 22 Banphaeo Dialysis Group (Bangkok), Banphaeo Hospital (Public organization), Bangkok, Thailand, 23 Division of Nephrology, Department of Internal Medicine, Faculty of Medicine, Prince of Songkla University, Songkhla, Thailand, 24 St. Michael's Hospital, Toronto, ON, Canada, 25 Division Nephrology, Faculty of Medicine Siriraj Hospital, Mahidol University, Bangkok, Thailand, 26 Division Nephrology, Faculty of Medicine, Ramathibodi Hospital, Mahidol University, Bangkok, Thailand

¶ Membership of the Advisory Board of Peritoneal Dialysis, Nephrology Society of Thailand is provided in the Acknowledgments
* Talerngsak.K@chula.ac.th

**Data Availability Statement:** All relevant data are within the paper and its Supporting Information files.

## Abstract

### Introduction

We sought to evaluate the predictors and outcomes of mold peritonitis in patients with peritoneal dialysis (PD).

### Methods

This cohort study included PD patients from the MycoPDICS database who had fungal peritonitis between July 2015-June 2020. Patient outcomes were analyzed by Kaplan Meier

**Funding:** This study was supported by the Thailand Science research and Innovation Fund Chulalongkorn University CU_FRB65_hea (19) _026_30_07, Chulalongkorn University, Thailand, the National Research Council of Thailand (156/ 2560), and Thailand Research Foundation (IRG5780017).

**Competing interests:** The authors have declared that no competing interests exist.

curves and the Log-rank test. Multivariable Cox proportional hazards model regression was used to estimating associations between fungal types and patients' outcomes.

## Results

The study included 304 fungal peritonitis episodes (yeasts n = 129, hyaline molds n = 122, non-hyaline molds n = 44, and mixed fungi n = 9) in 303 patients. Fungal infections were common during the wet season ($p$ <0.001). Mold peritonitis was significantly more frequent in patients with higher hemoglobin levels, presentations with catheter problems, and positive galactomannan (a fungal cell wall component) tests. Patient survival rates were lowest for non-hyaline mold peritonitis. A higher hazard of death was significantly associated with leaving the catheter in-situ (adjusted hazard ratio [HR] = 6.15, 95%confidence interval [CI]: 2.86–13.23) or delaying catheter removal after the diagnosis of fungal peritonitis (HR = 1.56, 95%CI: 1.00–2.44), as well as not receiving antifungal treatment (HR = 2.23, 95%CI: 1.25–4.01) or receiving it for less than 2 weeks (HR = 2.13, 95%CI: 1.33–3.43). Each additional day of antifungal therapy beyond the minimum 14-day duration was associated with a 2% lower risk of death (HR = 0.98, 95%CI: 0.95–0.999).

## Conclusion

Non-hyaline-mold peritonitis had worse survival. Longer duration and higher daily dosage of antifungal treatment were associated with better survival. Deviations from the 2016 ISPD Peritonitis Guideline recommendations concerning treatment duration and catheter removal timing were independently associated with higher mortality.

## Introduction

Fungal peritonitis is substantially more common in patients with peritoneal dialysis (PD) compared to the general population [1] and has high mortality ranging between 4.0 and 60.5% [2–16]. Retrospective studies and registries have demonstrated that fungal peritonitis accounts for between 1% and 24% of all peritonitis episodes in people undergoing PD, with absolute rates ranging from 0.01–0.09 episode/patient-year [2–17]. The highest prevalence has been reported in tropical countries, such as India, where the rates are as high as 24% (0.09 episodes/year) [7,9]. Although approximately 80 fungal species have been demonstrated as potentially causative organisms, *Candida* spp. is the predominant pathogen causing fungal peritonitis [2–11,13–16], whilst filamentous mold-associated peritonitis variably accounts for between 0% and 32% of fungal peritonitis episodes [2–4,7,9,11,14,16]. Clinically, both types of fungal peritonitis seem to manifest similarly in the literature. Therefore, the 2016 International Society for Peritoneal Dialysis (ISPD) Peritonitis Guidelines strongly recommend removing the PD catheter immediately after fungi are identified in PD patients with fungal peritonitis (1C), followed by a continuation of antifungal agents for an additional 2 more weeks (2C). No specific recommendations are made regarding the type and dose of antifungal medications to be administered [18]. However, these clinical speculation and recommendations are based mainly on studies of *Candida* peritonitis.

To date, only 3 retrospective studies have compared peritonitis outcomes between different causative fungal species. Wong et al. [4] demonstrated similar mortality rates between *Candida* and non-*Candida* species (47% vs. 37%). In contrast, Lo et al. [6] showed that *C. parapsilosis*

was associated with higher mortality than other *Candida* species. Chang et al. [11] found that *C. albicans* infection was a predictor of mortality compared to non-*Candida* fungi in univariate analysis but not in multivariable analysis. However, the results of previous studies should be interpreted with caution because of the comparatively small number of filamentous mold peritonitis episodes (less than 20 episodes) in each group. We leveraged the comprehensive Thailand Fungal PD-related Infectious Complications Surveillance (MycoPDICS) database to overcome these limitations. This surveillance registry was specifically designed to capture fastidious organisms causing PD-related infection, to conduct adequately powered, multivariable-adjusted comparisons of types of pathogens in PD patients with fungal peritonitis to better elucidate predictors and outcomes of mold peritonitis in patients with PD.

## Materials and methods

### Study design and population

This study included adult PD patients (≥18 years) recorded in the MycoPDICS database who had fungal peritonitis and catheter-related fungal infection between July 2015-June 2020. Written informed consent was obtained from all the participants or their legal substitute decision-makers. The study was approved by the Chulalongkorn University Institutional Review Board. To be eligible for inclusion, participants had to fulfill the 2016 ISPD Peritonitis Guidelines diagnostic criteria for peritonitis [18] by the presence of at least 2 of the following: 1) clinical symptoms of peritoneal inflammation, including abdominal pain and cloudy dialysate; 2) more than 100 leukocytes/mm$^3$ in PD effluent (PDE) with at least 50% neutrophils; 3) documentation of fungi in PDE or PD catheter by either smear or culture. The study was also registered in the Thai Clinical Trials Registry (Registration Number TCTR20210521009).

### Data collection

The MycoPDICS database is a national registry designed to survey the incidence of PD-related infections with fungus or environmental organisms under the Nephrology Society of Thailand (NST). It has been launched since the perceived burden of fungal peritonitis was high. The objectives of this surveillance include 1) monitoring fungal and environmental infections in PD patients, disease trends, and risk factors; 2) estimating the burden of disease; 3) providing pathogen isolates, detecting new pathogens, and monitoring for emerging antifungal resistance; and 4) supporting treatment platforms. The single case investigation involves 1) communication with the index patient and/or their caregivers using a semi-structured questionnaire on possible risk factors for infection and describing the household environment; 2) transportation of the suspected PD specimens to the central laboratory within 24 hours in an ice-shield container for organism identification, including PD bag, serum, and PD catheter (as needed). Identified species and antifungal agent susceptibility summaries are distributed back to the reporting units as soon as the results are available. The treatment regimen and schedule is under the attending physician's clinical judgment. 3) Collecting patient-level and facility-level information using a standard protocol and data collection instruments in all voluntary participating facilities. NST also undertakes a verification procedure through communication with the primary physicians and reference PD nurses to confirm the index case.

The retrieved data for this study included patient demographics, comorbidities at the start of dialysis, presenting symptoms and signs, presence of coexisting bacterial peritonitis, antibiotic use within the 3 months before fungal peritonitis, laboratory data at the onset of peritonitis, species of pathologic fungus and fungal characteristics, the initial and subsequent antimicrobial treatment, catheter removal and the time of removal, and patient outcomes.

## Definition

PD catheter malfunction was defined as mechanical failure to attain and maintain dialysate inflow or outflow sufficient to perform PD. Death related to fungal peritonitis was defined as the death of a patient with active peritonitis or sepsis secondary to peritonitis or within 4 weeks of initial diagnosis of fungal peritonitis [18]. According to wet smear, fungi were categorized into 3 groups: yeast, hyaline mold, and non-hyaline mold. For polymicrobial infections, fungal peritonitis was included if a fungus was at least one of the isolated organisms. Mixed fungal infection was defined as a concomitant infection with more than one species of fungus.

## Organism identification

At the central laboratory, 3 bottles of 50 mL of PDE obtained from the submitted PD bags were centrifuged at 3,500g for 15 minutes, and the supernatants were subsequently discarded. The remaining solution (around 5 mL) was mixed up with pellet and injected into bacterial and mycobacterial broths/agars to exclude concomitant bacterial/mycobacterial infection, including Bactec Plus Aerobic/F, BACTEC Plus Anaerobic/F vials (Dun Laoghaire, Ireland), BACTEC MGIT 960 media, Ogawa medium slants, blood agar, MacConkey agar (Oxoid, Basing-stoke, UK), Chocolate agar (Oxoid, Basing-stoke, UK), and specific agar plates (as needed) for 5–7 days (bacteria) and 2 months (mycobacteria) at 37˚C. For fungal culture, the pellet from another 50 mL of centrifuged PDE was streaked on Sabouraud dextrose agar (SDA) and specific agar plates (as needed), then incubated at 25˚C and 37˚C for 15–30 days. Yeast-form fungi were identified by API20c AUX kit (bioM´erieux, Marcy l'Etoile, France) based on biochemical reactions. Mold-form fungi were classified based on their sexual spores and conidia morphology.

Species were confirmed by molecular phylogeny using nucleotide sequences of internal transcribed spacer (ITS1/ITS4 primer; White et al., 1990) and large subunit region (5.8SR/LR7 primer; Vilgalys lab, Duke University) of the ribosomal RNA gene. The reaction mixture with fungal DNA was utilized as a positive control, and the reaction mixture without a template was used as a negative control. The experiments were repeated twice. The purified PCR products were then outsourced for Sanger sequencing service (First BASE Laboratories, Singapore Science Park II, Singapore). The sequencing results were subjected to BLASTN (National Center for Biotechnology Information Internet homepage) search against the GenBank database for homology identities. Antifungal susceptibility patterns of yeast and mold against common antifungal medications were assessed by Epsilometer test (bioMérieux, Marcy l' Etoile, France) and broth dilution technique (according to the CLSI document M38-A2 protocol), respectively.

## Statistical analysis

Results were expressed as frequencies and percentages for categorical variables and median and IQR for continuous variables. Differences between the three groups of patients were analyzed by the $\chi^2$-test and Fisher's exact test for categorical data and the Kruskal-Wallis test for continuous data. Only variables with missing less than 20% were included for analysis. After excluding mixed fungal infection, all primary patient outcomes were analyzed by survival analyses using Kaplan Meier curves together with the Log-rank test. Associations between variables and patient outcomes were first analyzed by univariate Cox proportional hazard regression. All variables with $p$ values of 0.20 or less were candidates for the multivariable Cox model with adjustment for age, gender, diabetes, employment status, PD vintage, hemoglobin, serum albumin, and high PDE leukocyte count. The assumption of proportional hazards was verified using Schoenfeld residuals and plots. Data were analyzed using the software packages Stata

16.1 (College Station, TX) and R 4.0.5 (R Core Team, Vienna). P-values less than 0.05 were considered statistically significant.

## Results and discussion

### Study population and distribution of fungal categories

Of 349 PD patients with fungal peritonitis (352 episodes) from 48 facilities in the MycoPDICS database, 305 episodes (304 participants) were fulfilled the criteria of diagnosis of fungal peritonitis. After excluding a case with an unknown patient outcome, 304 episodes in 303 participants were included in the current study (Fig 1). According to fungal morphology in wet smear, isolates included yeast (n = 129, 42%), hyaline mold (n = 122, 40%), non-hyaline mold (n = 44, 15%), and mixed fungi (n = 9, 3%). The most common yeast was *C. parapsilosis* (35%), followed by *C. tropicalis* (12%), and *C. albicans* (11%). Non-candida yeasts were found in 19% (n = 24). The three most common hyaline molds were *Aspergillus* (n = 52), *Fusarium* (n = 29), and *Penicillium* (n = 15). Dematiaceous mold (is termed if the fungal morphology or colonies appear as black or off-black colors) was the predominant non-hyaline mold (91%), including *Curvularia* (n = 15), *Cladosporium* (n = 8), *Exophiala* (n = 5), and *Bipolaris* (n = 3). A list of all fungal species isolated and categorized based on a wet smear is depicted in Table 1.

### Patient and clinical characteristics by fungal category

The mean age was 58 [49–66] years. The average PDE leukocyte count was 1,200 [400–3,100] cells/μL with neutrophil predominance (83 [70–91]%). Exposure to antibiotics within 3

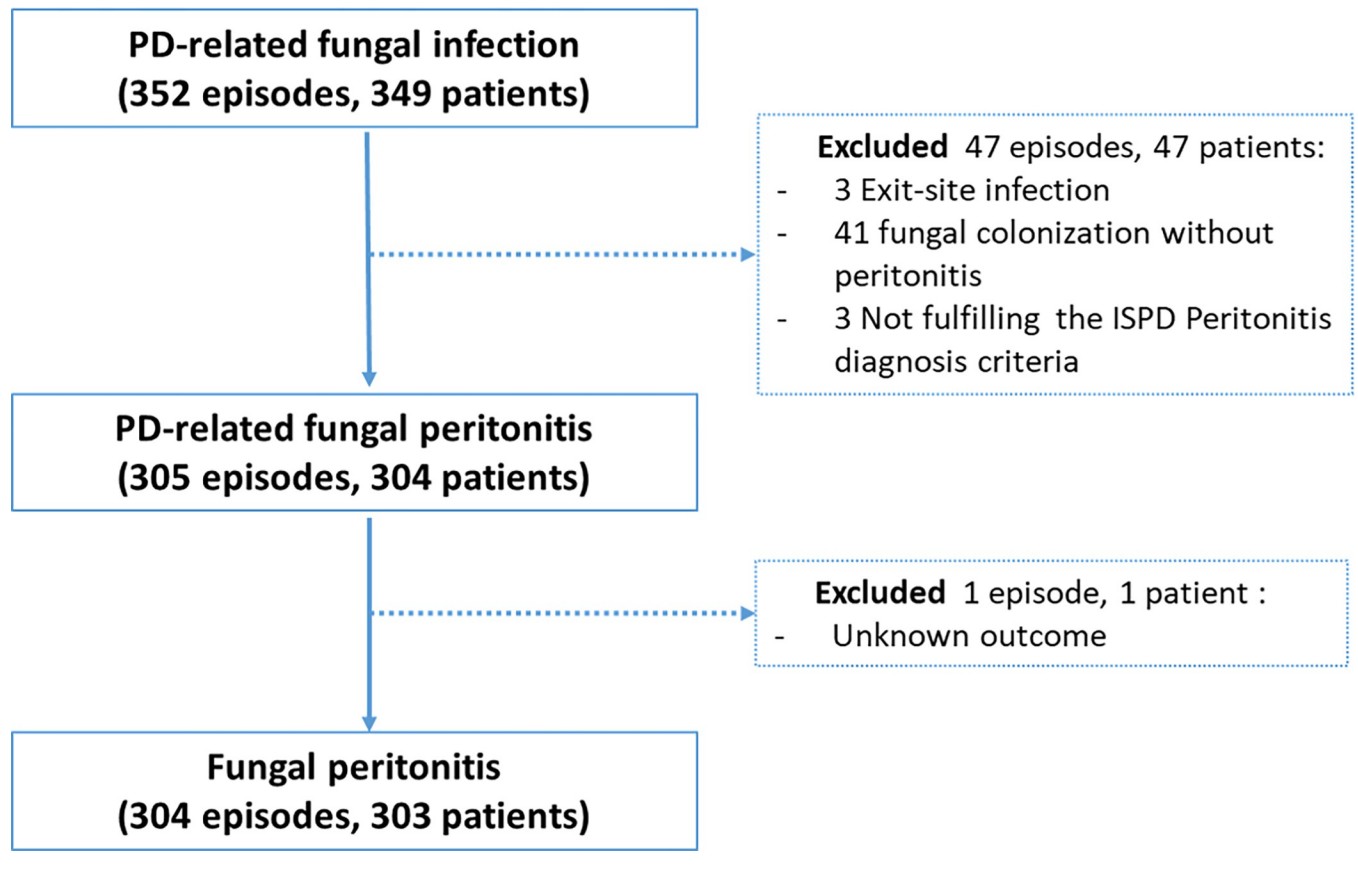

**Fig 1. Patient flow diagram.**

**Table 1. Spectrum of fungal organisms isolated from fungal peritonitis episodes.**

| Yeast (n = 129) | | Hyaline mold (n = 122) | | Non-hyaline mold (n = 44) | |
|---|---|---|---|---|---|
| • *Candida* (*C. parapsilosis*, 45; *C. tropicalis*, 15; *C. albicans*, 14; *C. guillermondii*, 10; *C. krusei*, 2; others, 11) | 105 | • *Aspergillus* (*A. flavus*, 30; *A. niger*, 6; *A. terreus*, 3; *A. fumigatus*, 3; *A. versicolor*, 2; Others 8) | 52 | • *Curvularia* (*C. lunata*, 8; *Curvularia* spp., 4; *C. hawaiiensis*, 2; *C. geniculate*, 1) | 15 |
| • *Trichosporon** (*T. asahii*, 6; other, 1) | 7 | • *Fusarium* (*F. solani*, 15; others 14) | 29 | • *Cladosporium* (*Cladosporium* spp., 6; *C. sphaerospermum*, 1; *C. tenuissimum*, 1) | 8 |
| • *Kodamaea* (*K. ohmeri*, 6) | 6 | • *Penicillium* (*P. citrinum*, 6; *P. rubens*, 1; Others 8) | 15 | • *Exophiala* (*Exophiala* spp., 2; *E. spinifera*, 2; *E. dermatitidis*, 1) | 5 |
| • *Blastobotrys* (*B. adeninivorans*, 3; *B. raffinosifermentans*, 2) | 5 | • *Paecilomyces* (*Paecilomyces* spp., 3; *P. formosus*, 2; *P. variotii*, 1) | 6 | • *Bipolaris* (*Bipolaris* spp., 3) | 3 |
| • *Cryptococcus** (*C. laurentii*, 1; other, 1) | 2 | • *Acremonium* (*A. implicatum*, 1; *A. obclavatum*, 1, Others, 3) | 5 | • *Exserohilum* (*Exserohilum* spp., 1; *E. rostratum*, 1) | 2 |
| • *Hyphopichia* (*H. burtonii*, 1) | 1 | • *Scedosporium* (*Scedosporium* spp., 2; *S. apiospermum*, 1) | 3 | • *Alternaria* (*A. alternate*, 1) | 1 |
| • *Rhodotorula** (*R. minuta*, 1) | 1 | • *Trichothecium* (*Trichothecium* spp., 2) | 2 | • *Cunninghamella*** (*C. bertholletiae*, 1; *C. echinulata*, 1) | 2 |
| • Others (2) | 1 | • Others (*C. intermedia*, 1) | 10 | • Others | 7 |

months prior to peritonitis was found in 42% of episodes. Of note, 8% of episodes (n = 24) had concomitant bacterial infection (*Acinetobacter baumannii*, 4; *Enterococcus faecium*, 3; others, 17). The median time from peritonitis onset to diagnosis of fungal peritonitis was 10 [6–17] days. Hypokalemia (defined as serum potassium <3.5 mEq/L) was found in 48% (n = 132) while hypoalbuminemia (defined as serum albumin <3.5 g/dL) was found in 83% (n = 226). Fungal infections—both yeasts and molds—were significantly more common during the wet season (Fig 2). Compared with yeast peritonitis, mold peritonitis was significantly more frequent in patients with higher hemoglobin levels, presentations with catheter malfunction or intraluminal colonization, and positive galactomannan (a fungal cell wall component that is shed by fungi during their growth and death) tests. Hyaline mold peritonitis had the lowest PD effluent (PDE) neutrophil percentage (Table 2).

## Treatment characteristics

Of interest, 11% (n = 33) and 13% (n = 40) of the fungal episodes did not receive PD catheter removal and antifungal medication, respectively. The average onset of PD catheter removal after the fungal diagnosis was 5 [2–10] days. The duration of antifungal therapy was 14 [14–16] days. Amphotericin B was the most common antifungal agent (70%), with an average dosage of 40 mg/day (0.7 mg/kg/day) and a duration of 14 days. It was combined with 5-flucytosine (5-FC) in 11 episodes (5%). Fluconazole, voriconazole, and itraconazole were used in 53 (17%), 25 (8%), and 8 (3%) episodes, respectively. Sequential therapy was prescribed in 32 episodes (11%), generally starting with amphotericin B and followed by fluconazole (n = 19), voriconazole (n = 9), or itraconazole (n = 4). Mold peritonitis was commonly treated with amphotericin B, whilst fluconazole was more frequently prescribed in yeast peritonitis (Table 2).

## Fungal peritonitis and subsequent death

The median follow-up time of the participants was 12.0 [3.6–26.2] months that there was no significant difference among groups. The all-cause mortality rates of fungal peritonitis at 1, 3, 6, and 12 months after diagnosis of peritonitis were 11%, 23%, 31%, and 36%, respectively. Using multivariable Cox proportional hazards model analysis, every year increase in age and

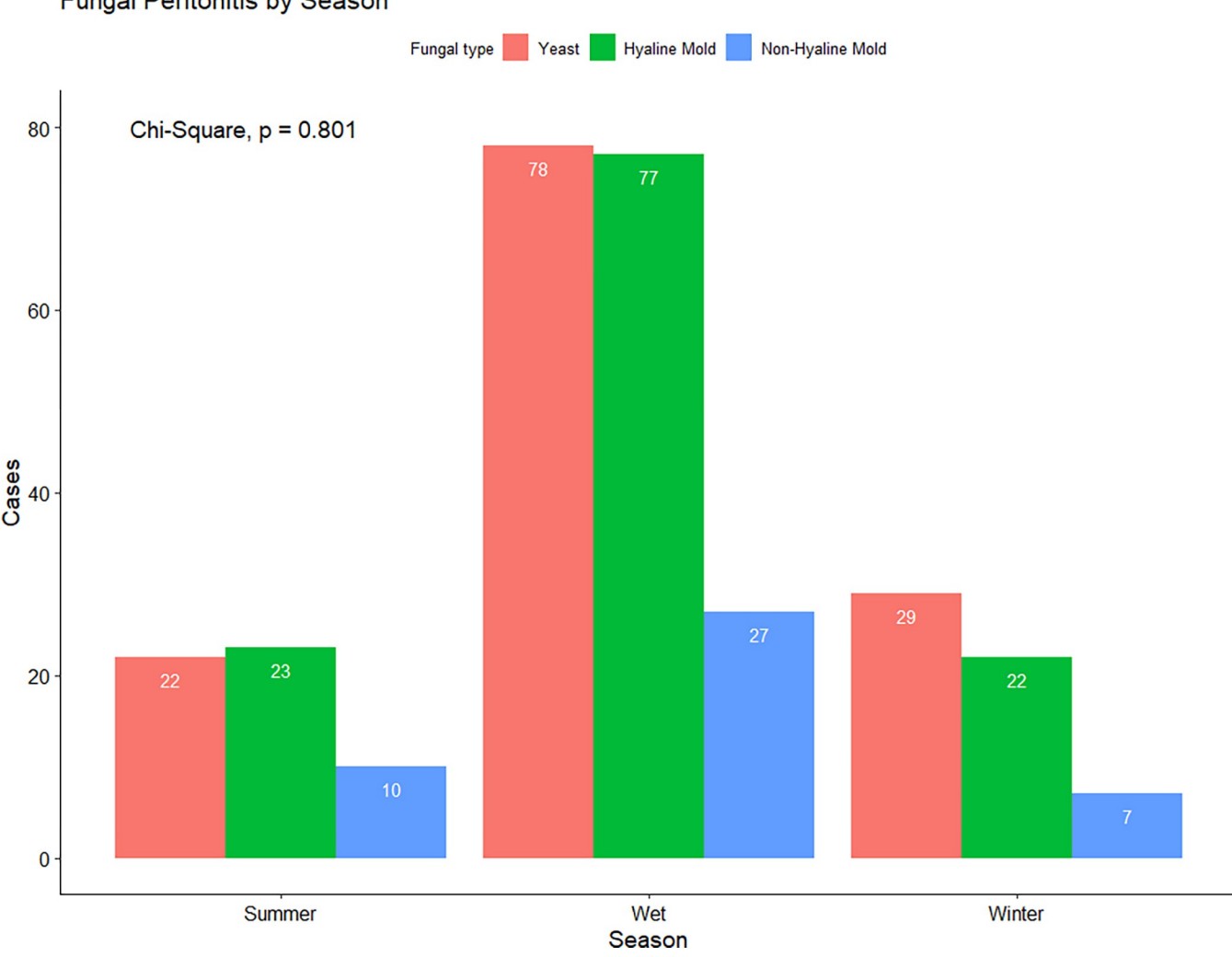

**Fig 2. Seasonal variation in the prevalence of fungal peritonitis episode according to fungal type by wet smear.**

PD vintage were associated with 2% and 18% higher risks of death (HR = 1.02, 95%CI:1.00–1.04 and HR = 1.18, 95%CI 1.07–1.30), whilst every 1 gm/dL decrease in serum albumin and hemoglobin levels at baseline were associated with 68% and 22% greater risks of mortality (HR = 1.68, 95%CI:1.17–2.42 and HR = 1.22, 95%CI:1.08–1.38). Presences of catheter problems (catheter colonization and malfunction) were associated with lower risks of death (HR = 0.55, 95%CI 0.33–0.89 and HR = 0.52, 95%CI:0.32–0.83). A higher hazard of death was significantly associated with leaving the catheter in situ (HR = 6.15, 95%CI:2.86–13.23) and delaying catheter removal after the diagnosis of fungal peritonitis (HR = 1.56, 95%CI:1.00–2.44). Mortality was also associated with not receiving antifungal treatment (HR = 2.23, 95% CI:1.25–4.01) and duration of antifungal treatment less than 2 weeks (HR = 2.13, 95%CI:1.33–3.43). Each additional day of antifungal therapy beyond the minimum 14-day duration was associated with a 2% reduction in the risk of death (HR = 0.98, 95%CI:0.95–0.999). Each 100 mg/day increase in total dosage of triazole was associated with an 18% lower risk of death (HR = 0.82, 95%CI 0.69–0.97)(Table 3). Kaplan Meier curves for patient survival in each category of fungal peritonitis are demonstrated in Fig 3.

**Table 2. Demographic, clinical and laboratory characteristics, and treatment of PD patients with fungal peritonitis according to fungal type.**

| Variables | Total (304) | Yeast (129) | Hyaline Mold (122) | Non-Hyaline Mold (44) | P value** |
|---|---|---|---|---|---|
| **Demographics** | | | | | |
| Age, years | 58 [49–66] | 58 [49–67] | 56 [47–65] | 60 [53–64] | 0.40[a] |
| Male gender, % | 154 (50.7) | 65 (50.4) | 61 (50.0) | 24 (54.6) | 0.87[b] |
| Diabetes, % | 130 (44.4) | 47 (37.6) | 54 (46.2) | 22 (52.4) | 0.18[b] |
| Employed, % | 169 (60.4) | 74 (63.3) | 67 (60.4) | 23 (53.5) | 0.53[b] |
| Automated PD, % | 5 (1.7) | 3 (2.4) | 2 (1.7) | 0 (0.0) | 0.85[c] |
| PD vintage, years | 2.0 [1.0–3.6] | 2.1 [1.0–3.9] | 1.9 [0.8–3.7] | 2.1 [1.1–4.4] | 0.66[a] |
| **Clinical characteristics** | | | | | |
| Peritonitis only, % | 112 (36.8) | 77 (59.7) | 25 (20.5) | 7 (15.9) | < 0.001[b] |
| Visible catheter colonization*, % | 189 (62.2) | 49 (38.0) | 98 (80.3) | 36 (81.8) | < 0.001[b] |
| Presence of catheter malfunction, % | 43 (14.1) | 2 (1.6) | 27 (22.1) | 11 (25.0) | < 0.001[b] |
| PDE leukocyte count, x1,000 cells/μL | 1.2 [0.4–3.1] | 1.2 [0.4–2.6] | 1.1 [0.4–4.1] | 1.1 [0.3–1.8] | 0.65[a] |
| PDE neutrophil, % | 83 [70–91] | 87 [75–93] | 80 [64–88] | 86 [70–92] | 0.004[a] |
| Preexisting exposure to antibiotics | 127 (41.8) | 56 (43.4) | 48 (39.3) | 20 (45.5) | 0.71[b] |
| Season*, % | | | | | 0.80[b] |
| Summer | 55 (18.1) | 22 (17.1) | 23 (18.9) | 10 (22.7) | |
| Wet | 189 (62.2) | 78 (60.5) | 77 (63.1) | 27 (61.4) | |
| Winter | 60 (19.7) | 29 (22.5) | 22 (18.0) | 7 (15.9) | |
| **Blood chemistries** | | | | | |
| Hemoglobin, g/dL | 9.4 [8.1–10.7] | 9.0 [7.6–10.4] | 9.5 [8.3–10.7] | 9.8 [8.9–11.0] | 0.03[a] |
| Leukocyte count, x1,000 cells/μL | 8.9 [6.6–11.4] | 8.8 [6.6–11.5] | 8.7 [6.3–11.2] | 9.1 [7.1–11.1] | 0.76[a] |
| Albumin, g/dL | 2.6 [2.1–3.2] | 2.6 [2.1–3.2] | 2.6 [2.1–3.4] | 2.8 [2.4–3.1] | 0.77[a] |
| Potassium, mEq/L | 3.5 [3.0–4.2] | 3.5 [3.0–4.3] | 3.5 [2.9–4.2] | 3.4 [3.0–4.3] | 0.88[a] |
| **Fungal characteristics** | | | | | |
| Bacterial coinfection, % | 24 (7.9) | 12 (9.3) | 6 (4.9) | 5 (11.4) | 0.27[b] |
| Positive galactomannan, % | 154 (71.3) | 57 (61.3) | 70 (79.6) | 22 (75.9) | 0.02[b] |
| Onset to fungal identification (from peritonitis), days | 10 [6–19] | 10 [7–19] | 11 [6–19] | 8 [4–15] | 0.37[a] |
| **PD catheter treatment** | | | | | |
| Receiving PDC removal, % | 271 (89.1) | 114 (88.4) | 110 (90.2) | 38 (86.4) | 0.77[b] |
| Onset to PDC removal (from peritonitis), days | 8 [4–17] | 7 [4–15] | 9 [5–19] | 7 [3–12] | 0.24[a] |
| Onset to PDC removal (from FP diagnosis), days | 5 [2–10] | 5 [3–12] | 5 [2–9] | 4 [1–8] | 0.20[a] |
| Immediate PDC removal (7 days from FP diagnosis)*, % | 173 (56.9) | 72 (55.8) | 71 (58.2) | 26 (59.1) | 0.90[b] |
| Antifungal therapy | | | | | |
| Receiving treatment, % | 264 (86.8) | 111 (86.1) | 111 (91.0) | 35 (79.6) | 0.14[b] |
| Duration of treatment, days | 14 [14–16] | 14 [10–17] | 14 [14–16] | 14 [7–15] | 0.08[a] |
| Adequate duration of treatment (14 days after PDC removal), % | 156 (51.3) | 69 (53.5) | 63 (51.6) | 19 (43.2) | 0.49[b] |
| Amphotericin B, % | 214 (70.4) | 81 (62.8) | 96 (78.7) | 31 (70.5) | 0.02[b] |
| Amphotericin B total dosage***, mg | 700 [500–700] | 700 [490–700] | 700 [595–700] | 700 [490–700] | 0.19[a] |
| Amphotericin B***, mg/day | 50 [40–50] | 50 [35–50] | 50 [40–50] | 50 [45–50] | 0.70[a] |
| Amphotericin B duration***, days | 14 [14–14] | 14 [14–14] | 14 [14–14] | 14 [14–14] | 0.07[a] |
| Fluconazole, % | 53 (17.4) | 37 (28.7) | 10 (8.2) | 5 (11.4) | < 0.001[b] |
| Voriconazole, % | 25 (8.2) | 7 (5.4) | 15 (12.3) | 2 (4.6) | 0.09[b] |
| Combination of antifungal agent*, % | 47 (15.6) | 15 (11.6) | 24 (19.8) | 7 (15.9) | 0.20[b] |

**Abbreviations:** FP, fungal peritonitis; PDC, peritoneal dialysis catheter, PDE, peritoneal dialysis effluent.

Categorical values are represented as number and percentages. Continuous values are represented as median [IQR].

*Total percentage is not equal to 100% due to rounding.

**Nine participants with mixed fungal peritonitis are excluded from analysis.

***Only patients who received Amphotericin B included.

[a]Kruskal-Wallis test.

[b]Pearson's χ2 test.

[c]Fisher's exact test.

**Table 3. Factors associated with mortality in patients with fungal peritonitis using univariable and multivariate analyses.**

| Variables | Unadjusted HR | 95% CI | *P* value | Adjusted HR model[a] | 95% CI | *P* value |
|---|---|---|---|---|---|---|
| Age per 1-year increment | 1.02 | 1.01–1.04 | 0.008 | 1.02 | 1.00–1.04 | 0.03 |
| Male gender | 1.06 | 0.72–1.56 | 0.76 | 1.28 | 0.82–2.01 | 0.28 |
| Diabetes | 1.25 | 0.84–1.84 | 0.27 | 1.33 | 0.82–2.15 | 0.25 |
| Employed state | 0.69 | 0.46–1.03 | 0.07 | 0.70 | 0.44–1.12 | 0.14 |
| Automated PD modality | 2.68 | 0.85–8.46 | 0.09 | 1.18 | 0.26–5.31 | 0.83 |
| PD vintage per 1-year increment | 1.08 | 1.00–1.17 | 0.05 | 1.18 | 1.07–1.30 | < 0.001 |
| **Clinical characteristics** | | | | | | |
| *Visible catheter colonization* | 0.66 | 0.44–0.97 | 0.03 | 0.55 | 0.33–0.89 | 0.01 |
| *Presence of catheter malfunction* | 0.62 | 0.42–0.91 | 0.02 | 0.52 | 0.32–0.83 | 0.006 |
| *PDE leukocyte count >1,090 cells/μL* | 1.11 | 0.75–1.63 | 0.61 | 1.17 | 0.75–1.83 | 0.49 |
| *PDE neutrophil count per 1% increment* | 1.01 | 1.00–1.02 | 0.08 | 1.01 | 1.00–1.02 | 0.11 |
| *Onset of FP diagnosis, 1 day incremental from first peritonitis date* | 1.01 | 1.00–1.02 | 0.11 | 1.01 | 0.99–1.02 | 0.35 |
| **Blood chemistries** | | | | | | |
| *Hemoglobin per 1 g/dL decrement* | 1.13 | 1.03–1.25 | 0.01 | 1.22 | 1.08–1.38 | 0.001 |
| *Albumin per 1 g/dL decrement* | 1.86 | 1.38–2.51 | < 0.001 | 1.68 | 1.17–2.42 | 0.005 |
| **Fungal characteristics (yeast as reference)** | | | | | | |
| *Hyaline mold* | 0.64 | 0.41–1.00 | 0.05 | 0.63 | 0.38–1.06 | 0.08 |
| *Non-Hyaline mold* | 1.31 | 0.79–2.19 | 0.30 | 1.21 | 0.70–2.16 | 0.52 |
| **PD catheter treatment** | | | | | | |
| *Not receiving PDC removal* | 3.20 | 1.73–5.93 | < 0.001 | 6.15 | 2.86–13.23 | < 0.001 |
| *Delay onset of PDC removal after FP diagnosis (>7 days)* | 1.73 | 1.17–2.55 | 0.006 | 1.56 | 1.00–2.44 | 0.049 |
| *Onset of PDC removal, 1 day incremental from date of FP diagnosis* | 1.01 | 1.00–1.03 | 0.08 | 1.01 | 1.00–1.03 | 0.14 |
| **Antifungal therapy** | | | | | | |
| *Not receiving treatment* | 3.67 | 2.38–5.66 | < 0.001 | 2.23 | 1.25–4.01 | 0.007 |
| *Inadequate duration of treatment (after PDC removal)* | 2.45 | 1.64–3.65 | < 0.001 | 2.13 | 1.33–3.43 | 0.002 |
| *Duration of antifungal treatment per 1-day increment from start of antifungal* | 0.97 | 0.94–0.99 | 0.003 | 0.98 | 0.95–0.999 | 0.04 |
| *Amphotericin B treatment* | 0.57 | 0.38–0.84 | 0.005 | 0.84 | 0.51–1.37 | 0.48 |
| *Amphotericin B dosage per 100 mg increment* | 0.93 | 0.88–0.98 | 0.004 | 0.96 | 0.90–1.01 | 0.13 |
| *Amphotericin B dosage per 10 mg/day increment* | 0.88 | 0.81–0.95 | 0.002 | 0.95 | 0.86–1.04 | 0.28 |
| *Amphotericin B duration per 1-day increment* | 0.97 | 0.95–0.99 | 0.03 | 0.98 | 0.96–1.01 | 0.24 |
| *Voriconazole treatment* | 0.39 | 0.14–1.06 | 0.07 | 0.40 | 0.13–1.29 | 0.13 |
| *Triazoles dosage per 1,000 mg increment* | 0.93 | 0.87–0.996 | 0.04 | 0.94 | 0.88–1.01 | 0.10 |
| *Triazole dosage per 100 mg/day increment* | 0.78 | 0.67–0.92 | 0.003 | 0.82 | 0.69–0.97 | 0.02 |
| *Triazole duration per 1-day increment* | 0.98 | 0.96–0.999 | 0.04 | 0.98 | 0.96–1.00 | 0.12 |
| *Combination of antifungal* | 0.66 | 0.36–1.20 | 0.17 | 0.59 | 0.28–1.24 | 0.16 |

**Abbreviations:** CI, confidence interval; FP, fungal peritonitis; HR, hazard ratio; PDC, peritoneal dialysis catheter, PDE, peritoneal dialysis effluent.

[a]Adjusted for age, gender, diabetes, employed state, PD vintage, hemoglobin, serum albumin, PDE leukocyte count > 1,090 cells/μL, and fungal type.

In sensitivity analyses comparing combination vs. single and 2 weeks vs. > 2–4 (>2, >3, or >4) weeks of antifungal therapy, treatment duration of > 4 weeks was associated with significantly lower mortality than a 2-week course in univariable analysis. However, the results were no longer statistically significant after applying multivariable adjustments (S1 Table). Combination and single antifungal regimens were comparable with respect to patient survival.

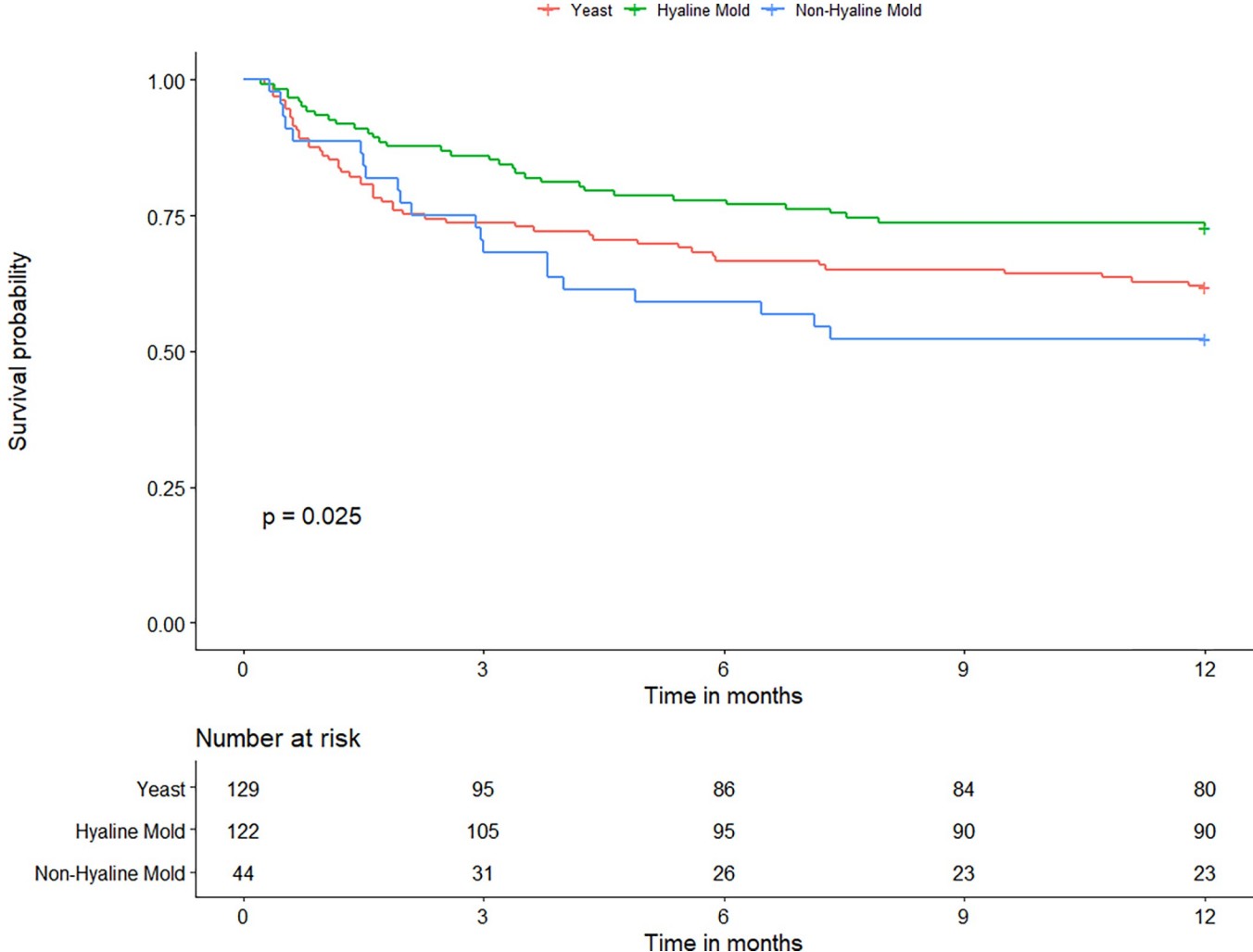

**Fig 3. Kaplan-Meier curves comparing patient survival according to different types of fungal peritonitis.**

The findings from this study differ somewhat from those of previous investigations. Specifically, the proportions of fungal peritonitis caused by hyaline molds (40%) and non-hyaline molds (15%) were relatively high, whilst the proportion due to *Candida* (35%) was relatively low compared with other published reports [2–17]. Generally, the reported prevalence of mold-associated peritonitis has varied from 0–32%, with the highest rate reported by the Australian and New Zealand Dialysis and Transplant (ANZDATA) Registry [10]. The reasons for the intriguingly high prevalence of mold peritonitis in our study are unclear but may have been related to climatic issues and the source of retrieved data used in this study. Thailand is located in a tropical area with hot and humid climates all year long, particularly in the wet season, usually accompanied by high precipitation rates and ambient temperatures. The contribution of climatic factors to the observed fungal peritonitis rates was supported by the high observed seasonal variations in fungal peritonitis, particularly during the wet season in both our study and the Australian study [19]. In the high-risk season, hot and humid climates may promote higher rates of skin perspiration, patient participation in outdoor activities, and the growth and virulence of environmental mold pathogens [19]. Additionally, our study involved surveillance registry data specifically designed to optimize capture of fastidious organisms,

including environmental molds, through specialized media, conditions, and techniques. This may have augmented detection of filamentous mold infections in peritonitis episodes in more recent years that would have previously been categorized as culture-negative [20].

Our study further demonstrated that mold peritonitis, both hyaline and non-hyaline molds, was significantly more frequent than yeast peritonitis in patients presenting with catheter problems. The explanation may involve fungal properties, growth, and characteristics. Generally, mold colonies are more easily recognized by their profuse growth of hyphae. In addition, they often exhibit larger variably-colored, with hair-like diffuse edges, and central foci, while yeast colonies are white to off-white smaller translucent with defined edges and no foci, which is similar to bacterial colonies [21–23]. These unique manifestations are intriguing because previous studies have not shown these distinctions, possibly related to lower observed numbers of mold peritonitis with reduced statistical power [2,8,9,11,15]. This distinct feature might alert the physician to suspicion of fungal peritonitis, resulting in an early prescription of anti-fungal medication and performing PD catheter removal. Therefore, the presence of catheter problems was a protective factor on patients' mortality in our study. Not surprisingly, a positive galactomannan test (a diagnostic marker for invasive fungal infections and PD-related peritonitis [24,25] and a prognostic marker for invasive aspergillosis in critical illness patients [26]) was detected less frequently in PDE from patients with yeast peritonitis (61%) than those with mold peritonitis (76–81%). Generally, yeasts contain a low amount of galactomannan in their cell walls, and *Pneumocystis jiroveci* and *Candida* spp. have absent galactomannan content in their cell walls. [24,25].

Although preexisting exposure to antibiotics was not different among 3 groups, almost half of each group (39–46%) had been received antibiotics in the past 3 months. Exposure to antibiotics has been known as one of the predictors of fungal peritonitis, particularly *Candida* infection. Unfortunately, only 5 episodes (1.5%) received antifungal prophylaxis, all of which were nystatin orally. According to the PDOPPS result, routine antifungal prophylaxis during antibiotic therapy varied considerably between countries, ranging from 7% in Japan and 23% in Thailand to 89% in Australia. However, most of the prophylaxis was employed in events with prolonged or broad-spectrum antibiotic uses, including peritonitis; the minority was used in all antibiotic courses, besides Australia [27]. Our finding supports the 2016 ISPD Peritonitis Guideline that "antifungal prophylaxis should be prescribed when PD patients receive all antibiotic courses to prevent fungal peritonitis (1B)," [18] particularly in countries with high prevalence fungal peritonitis.

Although the onset of fungal peritonitis diagnosis did not reach a statistical significance by both univariable and multivariable analyses, there were trends toward higher mortality in every extra day delay diagnosis of fungal peritonitis. Generally, the onset of fungal peritonitis is challenging to define in clinical practice, particularly in concomitant bacterial infection or secondary fungal infection cases. The fungus may colonize harmlessly inside the catheter. Nevertheless, if conditions are suitable, they can multiply and start to cause symptoms.

We found that a greater risk of death was associated with patients with older age, longer PD vintage, anemia, and hypoalbuminemia. Hypoalbuminemia has been revealed as a predictor for mortality in fungal peritonitis by Ram et al [9]. and as a negligible predictor by the other groups [11,14]. The previous studies did not find PD vintage and anemia to be risks factors of mortality in fungal peritonitis patients [11,14]. However, Nadeau-Fredette et al. found a trend of longer PD vintage in non-survived patients (2.5 and 5.0 years) [14]. These disparities are likely attributed to a small number of deaths in previous studies [9,11,14]. The results might be imprecise as the analyses were based on less than 30 deaths, whereas our analysis incorporated 70 (at 3 months) and 94 deaths (at 6 months). The sample size limitation might mask other

possible interactions and associations among patient characteristics, laboratories, and clinical outcomes.

This study strongly supported the 2016 ISPD Peritonitis Guidelines' recommendation for "immediate catheter removal when fungi are identified in PD effluent (1C)" [18] by demonstrating that leaving the catheter in situ or delaying catheter removal after the diagnosis of fungal peritonitis was strongly and independently associated with a higher mortality rate. A similar finding has been reported by previous studies [4,9,11]. Ram et al. [9] demonstrated that the mortality rate increases exponentially with increasingly delayed onset of the catheter removal, 19% (1 day), 67% (1 week), and 94% (1 month). However, the definition of delay onset used was varied across the literature [2–17]. Attempting to treat fungal peritonitis with the catheter in situ might leave an ongoing source of infection and impair the effectiveness of antifungals.

Of note, 11% of the fungal episodes in our study did not receive PD catheter removal despite the strong ISPD recommendation [18] of catheter removal in fungal peritonitis episodes soon after diagnosis. This finding may have reflected a lack of clinician appreciation of the virulence of mold peritonitis in Thailand. Since most Thai PD facilities (91%) have limited ability to culture filamentous mold [28], the attending nephrologists may also have had limited experience treating fungal peritonitis and limited access to infectious disease specialists. Moreover, some facilities might have had limited facility HD backup support, resulting in a reluctance to remove the PD catheter and subsequent deviation in practice from the ISPD Guideline recommendation. This will require further exploration.

Our study also found that longer durations of antifungal treatment beyond 2 weeks and higher dosages were associated with lower mortality rates. These findings support and extend the 2016 ISPD Peritonitis Guidelines recommendation that "treatment with an appropriate antifungal agent be continued for at least 2 weeks after catheter removal (2C). "[18] The optimal duration, dose, and choice of antifungal agents have not been previously established, probably due to previous studies' sparsity and small sample sizes [18,29–31]. Our findings would guide the clinicians' treatment decisions and prevent deleterious outcomes. A future revision of the guidelines concerning the finding of 2% decrease in mortality for every extra day beyond the minimum 14-day duration of antifungal medication is warranted.

The strengths of this study include its long length of follow-up (median 12.0, IQR 3.6–26.2 months), large sample size (304 fungal peritonitis episodes), and high cumulative number of death events (109 cases in 1 year), which helped to augment statistical power. However, some limitations also need to be highlighted. Firstly, participation in the surveillance registry was voluntary and not subjected to external audits. Consequently, the possibilities of ascertainment biases cannot be excluded. Despite adjusting for several demographic and clinical factors, the possibility of residual confounding also cannot be excluded. Secondly, the high detection rate of environmental mold raises a concern of specimen contamination during specimen sampling and handling in the registry data. However, the surveillance registry was well conducted, with orientation provided at all sites collecting and handling the specimens with strict aseptic technique. Of interest, colonization of the fungus was observed inside the catheter collected from most cases with filamentous fungus (80–82%), and subsequently, cultivation of the removed catheter confirmed its presence, thereby supporting its role as a genuine pathogen. In addition, the fungal cell wall in the PDE was tested in some cases to confirm the true positives. Finally, the observational design of this study means that causal inferences cannot be drawn.

## Conclusions

In conclusion, mold peritonitis was more frequently associated with higher hemoglobin levels, presentations with catheter malfunction or intraluminal colonization, and a positive

galactomannan test. Non-hyaline mold peritonitis was associated with the worst survival rates. Deviations from the 2016 ISPD Peritonitis Guideline recommendations concerning treatment duration and catheter removal timing were independently associated with higher mortality. Longer duration and higher daily dosage of antifungals were associated with lower mortality. Further investigation to identify more effective interventions that are specific to fungus type is warranted.

## Supporting information

**S1 Fig. Kaplan-Meier curves comparing patient survival following treatment of fungal peritonitis with and without PD catheter removal.**
(DOCX)

**S1 Table. Sensitivity analyses of duration of antifungal therapies and mortality among patients with fungal peritonitis using univariable and multivariate analyses.**
(DOCX)

**S1 File.**
(DOCX)

## Acknowledgments

We would like to acknowledge the contributions of the members of the Advisory Board of PD, NST, who are not listed as the authors, including Anutra Chittinandana, MD and Duangkamol Wongsawan, MD, Bhumibol Adulyadej Hospital; Chanchana Boonyakrai, MD, Taksin Hospital; Dhavee Siriwong, MD, Khon Kaen University; Monchai Siribamrungwong, MD, Lerdsin Hospital; Pornchai Kingwatanakul, MD, Department of Pediatrics, Chulalongkorn University; Solos Jaturapisanukul, MD, Vajira Hospital; Somchai Yongsiri, MD, Burapha University; Surapong Narenpitak, MD, Udonthani Hospital; Tanawoot Limlek, MD, Krabi Hospital; Thanee Eiamsitrakoon, M.D., Chulabhorn international College Medicine, Thammasat University; Yuttitham Suteeka, MD, Chiangmai University. Sarinya Boongird, MD, Department of Medicine, Ramathibodi Hospital, Mahidol University; Kamol Khositrangsikun, MD, Department of Internal Medicine, Maharaj Nakhon Sri Thammarat Hospital; and Laddaporn Wongluechai, MD, Department of Internal Medicine, Maharat Nakhon Ratchasima Hospital. We grateful thank to the staffs, nurses, and all investigators who work at reporting centers, including Ms. Piyaporn Towannang, King Chulalongkorn Memorial Hospital; Ms. Nisa Thongbor, Ms. Sirinart Raweewan, and Ms.Jitta Matawon, Sunpasitthiprasong Hospital; and Ms. Nipa Aiyasanon, Siriraj Hospital; Ms.Donkum Kaewboonsert, and Ms. Pensri Uttayotha, Phayao Hospital; Wichai Sopassathit, MD, Kittisak Tangjittrong, MD, and Ms. Salakjit Pitakmongkol, Pranangklao Hospita; Ms. Bunpring Jaroenpattrawut, Nakhon Pathom Hospital; Somphon Buranaosot, MD, Sukit Nilvarangkul, MD, and Ms. Warakoan Satitkan, Bangkok Metropolitan Administration General Hospital; Wanida Somboonsilp, MD, Pimpong Wongtrakul, MD, Ms. Ampai Tongpliw, and Ms. Anocha Pullboon, Chaoprayayomraj Hospital; Chanchana Boonyakrai, MD, and Ms. Montha Jankramol, Taksin Hospital; Surapong Narenpitak, MD, Mrs. Apinya Wechpradit, and Ms.Wannaporn Uthaiwat, Udonthani Hospital; Ms. Chadarat Kleebchaiyaphum, Chaiyaphum Hospital, Ms.Worauma Panya, and Ms. Siriwan Thaweekote, Mukdahan Hospital; Sriphrae Uppamai, MD, and Ms. Sirirat Sirinual, Sukhothai Hospital; Puntapong Taruangsri, MD, Setthapon Panyatong, MD, Ms. Boontita Prasertkul, and Ms. Thanchanok Buanet, Nakornping Hospital; Ms. Panthira Passorn, Sawanpracharak Hospital; Niwat Lowmseng, MD and Ms. Rujira Luksanaprom, Trang Hospital; Angsuwarin Wongpiang, MD, and Ms. Metinee Chaiwut, Pong Hospital; Ms. Ruchdaporn Phaichan, Chaophraya

Abhaibhubejhr Hospital; Peerapach Rattanasoonton, MD, and Ms. Wanlaya Thongsiw, Trat Hospital; Narumon Lukrat, MD, and Ms. Sayumporn Thaitrng, KhueangNai Hospital; Phichit Songviriyavithaya, MD, Ms. Yupha Laoong, and Ms. Niparat Pikul, Amnatcharoen Hospital; Ms. Navarat Rukchart, Ms.Korawee Sukmee, and Ms. Wandee Chantarungsri, Songkhla Hospital; Kamol Khositrangsikun, MD, Maharaj Nakhon Sri Thammarat Hospital; Ms. Sureewan Ratanakitsunthorn, Phra Nakhon Si Ayutthaya Hospital; Puttinan Namdee, MD, and Ms. Nipa nonbunta, Lomsak Hospital; Rhonachai Lawsuwanaku, MD, and Mrs. Wacharee Rattanawong, Chonburi Hospital; Piyanut Pratipannawat, MD, Kalasin Hospital; Suwattanachai Nurnuansuwan, MD, and Major. Nipaporn Sanorklang, Fort Suranari Hospital; Patchara Tanateerapong, MD, Kamonrat Chongthanakorn, MD, Mrs. Patchara Assawaboonyalert, and Ms. Julaluk Wongnaya, Charoenkrung Pracharak Hospital; Veerapatr Nimkietkajorn, MD, and Ms. Pasunun Keawsinark, Buddhachinaraj hospital; Soontorn Pinpaiboon, MD, and Mrs. Chantana Tongchuen, Kamphaengphet Hospital; Ms. Numpueng Jiranunsakul, Jainad Narendra Hospital; Theerapon Sukmark, MD, and Ms. Juntana Boonchoo, Thungsong Hospital; Sumonkarn Lapkitichaloenchai, MD, Nopparat Rajathanee Hospital; Poonlarb Panjaluk, MD, and Mrs. Onnitcha Jankhum, Banglamung Hospital; Mananya Wanpaisitkul, MD, and Ms. Chalearmsri Marod, Banpong Hospital; Pattanasak Thangnak, MD, and Mrs. Melanee Saengplaeng, Benchalak Community Hospital Commemorating His Majesty the King' 80th Birthday Anniversary; Thawat Tiawilai, MD, Photharam Hospital, Ms. Rossukon Tantivichitvej, and Mrs.Rapeephan Chantarasorn, Photharam Hospital; Mrs.Pattarasri Pimta, Mahasarakham Hospital; Jidapa Mahamongkhonsawat, MD, and Mrs.Supanee Wongsawat, Sichon Hospital; Laddaporn Wongluechai, MD, Maharat Nakhon Ratchasima Hospital, Thailand.

## Author Contributions

**Conceptualization:** Talerngsak Kanjanabuch.

**Data curation:** Talerngsak Kanjanabuch, Tanawin Nopsopon, Krit Pongpirul.

**Formal analysis:** Tanawin Nopsopon, Krit Pongpirul.

**Funding acquisition:** Talerngsak Kanjanabuch.

**Investigation:** Talerngsak Kanjanabuch, Tanittha Chatsuwan, Nibondh Udomsantisuk.

**Methodology:** Tanawin Nopsopon.

**Project administration:** Talerngsak Kanjanabuch.

**Validation:** Talerngsak Kanjanabuch.

**Visualization:** Talerngsak Kanjanabuch.

**Writing – original draft:** Talerngsak Kanjanabuch.

**Writing – review & editing:** Talerngsak Kanjanabuch, Tanittha Chatsuwan, Sirirat Purisinsith, David W Johnson, Nibondh Udomsantisuk, Guttiga Halue, Pichet Lorvinitnun, Pongpratch Puapatanakul, Krit Pongpirul, Ussanee Poonvivatchaikarn, Sajja Tatiyanupanwong, Saowalak Chowpontong, Rutchanee Chieochanthanakij, Oranan Thamvichitkul, Worapot Treamtrakanpon, Wadsamon Saikong, Uraiwan Parinyasiri, Piyatida Chuengsaman, Phongsak Dandecha, Jeffrey Perl, Kriang Tungsanga, Somchai Eiam-Ong, Suchai Sritippayawan, Surasak Kantachuvesiri.

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
