## [Decision Letter · Decision Letter 0]

31 Jan 2022

PONE-D-21-30246Predictors and Outcomes of PD-Related Infections due to Filamentous MoldsPLOS ONE

Dear Dr. Kanjanabuch,

Thank you for submitting your manuscript to PLOS ONE. After careful consideration, we feel that it has merit but does not fully meet PLOS ONE’s publication criteria as it currently stands. Therefore, we invite you to submit a revised version of the manuscript that addresses the points raised during the review process.

We look forward to receiving your revised manuscript.

Kind regards,

Vivekanand Jha

Academic Editor

PLOS ONE

Journal Requirements:

2. Please provide additional details regarding participant consent. In the Methods section, please ensure that you have specified (1) whether consent was informed and (2) what type you obtained (for instance, written or verbal). If your study included minors, state whether you obtained consent from parents or guardians. If the need for consent was waived by the ethics committee, please include this information.

4. Thank you for stating the following in the Acknowledgments/ Disclosure Section of your manuscript: 

TK has received consultancy fees from VISTERRA as a country investigator and current recipient of the National Research Council of Thailand and received speaking honoraria from Astra Zeneca and Baxter Healthcare. JP has received speaking honoraria from Astra Zeneca, Baxter Healthcare, DaVita Healthcare Partners, Fresenius Medical Care, Dialysis Clinics Incorporated, Satellite Healthcare, and served as a Baxter consultant Healthcare, DaVita Healthcare Partners, Fresenius Medical Care, and LiberDi. DJ has previously received consultancy fees, research grants, speaker's honoraria, travel sponsorships from Baxter Healthcare and Fresenius Medical Care, consultancy fees from AstraZeneca, Bayer, and AWAK and speaker's honoraria from Ono and BI & Lilly, and travel sponsorships from Amgen. He is also supported by an Australian National Health and Medical Research Council (NHMRC) Leadership Investigator Grant. All other authors have no financial conflicts of interest to declare. All funders and pharmaceutical companies did not have any role in the study design, data collection, data analysis, data reporting, and the decision to submit for publication. This study was supported by Rachadaphiseksompot Endorsement Fund (CU-GRS_60_12_30_05), Chulalongkorn University, National Research Council of Thailand (156/2560), and Thailand Research Foundation (IRG5780017).

TK has received consultancy fees from VISTERRA as a country investigator and current recipient of the National Research Council of Thailand and received speaking honoraria from Astra Zeneca and Baxter Healthcare. JP has received speaking honoraria from Astra Zeneca, Baxter Healthcare, DaVita Healthcare Partners, Fresenius Medical Care, Dialysis Clinics Incorporated, Satellite Healthcare, and served as a Baxter consultant Healthcare, DaVita Healthcare Partners, Fresenius Medical Care, and LiberDi. DJ has previously received consultancy fees, research grants, speaker's honoraria, travel sponsorships from Baxter Healthcare and Fresenius Medical Care, consultancy fees from AstraZeneca, Bayer, and AWAK and speaker's honoraria from Ono and BI & Lilly, and travel sponsorships from Amgen. He is also supported by an Australian National Health and Medical Research Council (NHMRC) Leadership Investigator Grant. All other authors have no financial conflicts of interest to declare. All funders and pharmaceutical companies did not have any role in the study design, data collection, data analysis, data reporting, and the decision to submit for publication. This study was supported by Rachadaphiseksompot Endorsement Fund (CU-GRS_60_12_30_05), Chulalongkorn University, National Research Council of Thailand (156/2560), and Thailand Research Foundation (IRG5780017).

5. One of the noted authors is a group or consortium [Nephrology Society of Thailand]. In addition to naming the author group, please list the individual authors and affiliations within this group in the acknowledgments section of your manuscript. Please also indicate clearly a lead author for this group along with a contact email address.

Reviewers' comments:

Reviewer's Responses to Questions

**Comments to the Author**

1. Is the manuscript technically sound, and do the data support the conclusions?

Reviewer #1: Yes

Reviewer #2: Yes

2. Has the statistical analysis been performed appropriately and rigorously? 

Reviewer #1: Yes

Reviewer #2: Yes

3. Have the authors made all data underlying the findings in their manuscript fully available?

Reviewer #1: Yes

Reviewer #2: Yes

4. Is the manuscript presented in an intelligible fashion and written in standard English?

Reviewer #1: Yes

Reviewer #2: Yes

5. Review Comments to the Author

Reviewer #1: Fungal Peritonitis is a dreaded complication of CAPD associated with technique failure, catheter removal, morbidity and mortality. This is a Multi centric study from Thailand whose experience with CAPD is huge given its PD First Policy. Multiple Risk factors were analysed well and my suggestions :

1. In the Materials and Methods section, a paragraph should be added about the specific specialised techniques used in isolation of these fastidious fungal pathogens. There is a passing mention about it in the Discussion section . But more details are required so that it can be corroborated by other centres

2. In the Discussion section , the first para can be omitted since it is a repetition of Results .

3. As a Recommendation based on this Paper do the Authors feel that additional week or two of anti fungal therapy will improve the survival? If they feel so, then it can be added as a recommendation

Reviewer #2: The study “Predictors and Outcomes of PD-Related Infections due to Filamentous Molds” is an interesting study. This cohort study included PD patients from the MycoPDICS database of Thailand who had fungal peritonitis between July 2015-June 2020. Authors conclude that Non-hyaline-mold peritonitis had worse survival. Longer duration and higher daily dosage of antifungal treatment were associated with better survival.

Comments:

1. The study included 304 fungal peritonitis episodes (yeasts n=133, hyaline molds n=122, and non-hyaline molds n=44) in 303 patients. A total sum of all episodes 133+122+44=299, not 304.

2. Authors have described the causative organism and outcomes of fungal peritonitis episodes only; however, it would be of interest to know the overall peritonitis rate and fungal peritonitis rate in that cohort of patients.

3. Authors have written that 11% (n=33) and 13% (n=40) of the fungal episodes did not receive PD catheter removal and antifungal medication, respectively. Overall, this percentage is a large number, can the authors mention that despite the recommendation of catheter removal in fungal peritonitis episodes soon after diagnosis, why catheters were not removed and antifungals were not given.

4. How many patients were treated with anti-fungal with catheters in situ and their success if any?

5. Each additional day of antifungal therapy beyond the minimum 14-day duration was associated with a 2% reduction in the risk of death (HR=0.98, 95%CI:0.95-0.999). Can authors identify, why these patients’ received antifungals beyond 14 days? was It a persistent ongoing fungal infection or something else? Can authors suggest an optimal duration of anti-fungal therapy?

6. Anti-fungal choices varied? Only 70% received Amphotericin- B, which was the basis of choosing other antifungals and combinations.

The only strength of the article is a large number of fungal episodes, and mortality predictors. Authors need to address the above mentioned comments to improve the manuscript.

6. PLOS authors have the option to publish the peer review history of their article (what does this mean?). If published, this will include your full peer review and any attached files.

Reviewer #1: **Yes: **Krishnaswamy Sampathkumar

Reviewer #2: No

---

## [Author Response · Author response to Decision Letter 0]

6 Apr 2022

Responses to Comments from Editor and Reviewers:

Reviewer #1 

Fungal Peritonitis is a dreaded complication of CAPD associated with technique failure, catheter removal, morbidity and mortality. This is a Multi centric study from Thailand whose experience with CAPD is huge given its PD First Policy. Multiple Risk factors were analysed well and my suggestions :

1) In the Materials and Methods section, a paragraph should be added about the specific specialised techniques used in isolation of these fastidious fungal pathogens. There is a passing mention about it in the Discussion section . But more details are required so that it can be corroborated by other centres

 Response: A paragraph about the specific specialised techniques used in the isolation of these fastidious fungal pathogens has been added to the MATERIALS AND METHODS section accordingly, as below.

 “At the central laboratory, 3 bottles of 50 mL of PDE obtained from the submitted PD bags were centrifuged at 3,500g for 15 minutes, and the supernatants were subsequently discarded. The remaining solution (around 5 mL) was mixed up with pellet and injected into bacterial and mycobacterial broths/agars to exclude concomitant bacterial/mycobacterial infection, including Bactec Plus Aerobic/F, BACTEC Plus Anaerobic/F vials (Dun Laoghaire, Ireland), BACTEC MGIT 960 media, Ogawa medium slants, blood agar, MacConkey agar (Oxoid, Basing-stoke, UK), Chocolate agar (Oxoid, Basing-stoke, UK), and specific agar plates (as needed) for 5-7 days (bacteria) and 2 months (mycobacteria) at 37°C. For fungal culture, the pellet from another 50 mL of centrifuged PDE was streaked on Sabouraud dextrose agar (SDA) and specific agar plates (as needed), then incubated at 25°C and 37°C for 15-30 days. Yeast-form fungi were identified by API20c AUX kit (bioM´erieux, Marcy l’Etoile, France) based on biochemical reactions. Mold-form fungi were classified based on their sexual spores and conidia morphology. 

 Species were confirmed by molecular phylogeny using nucleotide sequences of internal transcribed spacer (ITS1/ITS4 primer; White et al., 1990) and large subunit region (5.8SR/LR7 primer; Vilgalys lab, Duke University) of the ribosomal RNA gene. The reaction mixture with fungal DNA was utilized as a positive control, and the reaction mixture without a template was used as a negative control. The experiments were repeated twice. The purified PCR products were then outsourced to Sanger sequencing service (First BASE Laboratories, Singapore Science Park II, Singapore). The sequencing results were subjected to BLASTN (National Center for Biotechnology Information Internet homepage) search against the GenBank database for homology identities. Antifungal susceptibility patterns of yeast and mold against common antifungal medications were assessed by Epsilometer test (bioMérieux, Marcy l’ Etoile, France) and broth dilution technique (according to the CLSI document M38-A2 protocol), respectively.” 

2) In the Discussion section , the first para can be omitted since it is a repetition of Results.

 Response: The first paragraph of the DISCUSSION section has been omitted accordingly. 

3) As a Recommendation based on this Paper do the Authors feel that additional week or two of anti fungal therapy will improve the survival? If they feel so, then it can be added as a recommendation.

 Response: The power of this study is not adequate to address this issue. The sensitivity analyses of 2 vs. > 2-4 (>2, >3, or >4) weeks and single vs. combination of antifungal therapy are shown in S1 Table and have been added to the RESULTS AND DISCUSSION section, as below. 

“In sensitivity analyses comparing combination vs. single and 2 weeks vs. > 2-4 (>2, >3, or >4) weeks of antifungal therapy, treatment duration of > 4 weeks was associated with a significantly lower mortality than a 2-week course in univariable analysis. However, the results were no longer statistically significant after applying multivariable adjustments (S1 Table). Combination and single antifungal regimens were comparable with respect to patient survival.”

Reviewer #2

The study “Predictors and Outcomes of PD-Related Infections due to Filamentous Molds” is an interesting study. This cohort study included PD patients from the MycoPDICS database of Thailand who had fungal peritonitis between July 2015-June 2020. Authors conclude that Non-hyaline-mold peritonitis had worse survival. Longer duration and higher daily dosage of antifungal treatment were associated with better survival.

1) The study included 304 fungal peritonitis episodes (yeasts n=133, hyaline molds n=122, and non-hyaline molds n=44) in 303 patients. A total sum of all episodes 133+122+44=299, not 304.

 Response: We sincerely apologize for the typographical errors in the ABSTRACT section. According to fungal morphology in wet smear, isolates included yeast (n=129, 42%), hyaline mold (n=122, 40%), non-hyaline mold (n=44, 15%), and mixed fungi (n=9, 3%). The ABSTRACT section has been revised accordingly as below.

 “The study included 304 fungal peritonitis episodes (yeasts n=129, hyaline molds n=122, non-hyaline molds n=44, and mixed fungi n=9) in 303 patients.”

2) Authors have described the causative organism and outcomes of fungal peritonitis episodes only; however, it would be of interest to know the overall peritonitis rate and fungal peritonitis rate in that cohort of patients.

 Response: Since the registry is voluntary, not all peritonitis episodes were submitted and reported to it. However, almost half of the reported centers (22 out of 48 centers) were participating in the Peritoneal Dialysis Outcomes and Practice Patterns Study (PDOPPS), a large prospective international cohort study in PD, in collaboration with the International Society for Peritoneal Dialysis (ISPD), which has recruited participants from many countries, including Thailand. We, therefore, have calculated the total and fungal peritonitis rates from the Thailand PDOPPS data during the same period. We found that the overall and fungal peritonitis rates were 0.24 and 0.02 episodes/patient-year, respectively.

3) Authors have written that 11% (n=33) and 13% (n=40) of the fungal episodes did not receive PD catheter removal and antifungal medication, respectively. Overall, this percentage is a large number, can the authors mention that despite the recommendation of catheter removal in fungal peritonitis episodes soon after diagnosis, why catheters were not removed and antifungals were not given.

 Response: The uptake of the 2016 ISPD Peritonitis Guidelines recommendation [18] is low in Thailand, particularly in the rural and provincial areas. According to the PDOPPS study, center adherence to this guideline was generally suboptimal in Thailand. Example of such a practice deviations are that fungal prophylaxis during all antibiotic courses and exit-site prophylaxis were used in only 23% and 21% of facilities, despite this being level 1B and 1A recommendations by the ISPD guidelines, respectively [Boudville N et al. NDT 2019]. The following text has been added to the DISCUSSION section.

“Of note, 11% of the fungal episodes in our study did not receive PD catheter removal despite the strong ISPD recommendation [18] of catheter removal in fungal peritonitis episodes soon after diagnosis. This finding may have reflected a lack of clinician appreciation of the virulence of mold peritonitis in Thailand. Since most Thai PD facilities (91%) have limited ability to culture filamentous mold [28], the attending nephrologists may also have had limited experience treating fungal peritonitis and limited access to infectious disease specialists. Moreover, some facilities might have had limited facility HD backup support, resulting in a reluctance to remove the PD catheter and subsequent deviation in practice from the ISPD Guideline recommendation. This will require further exploration.”

4) How many patients were treated with antifungal with catheters in situ and their success if any?

 Response: In 264 participants who received antifungal medication, there were 27 PD catheters left in-situ. The participants with PD catheters left in-situ tended to have a higher mortality at 3-month follow-up than participants who had their PD catheters removed (33% vs. 18%, p = 0.052). Ram et al. [19] demonstrated that the mortality rate increases exponentially with increasingly delayed onset of catheter removal: 19% (1 day), 67% (1 week), and 94% (1 month) [19]. Therefore, attempting to treat fungal peritonitis with the catheter in-situ might leave an ongoing source of infection and impair the effectiveness of antifungals, such that it should be discouraged. 

5) Each additional day of antifungal therapy beyond the minimum 14-day duration was associated with a 2% reduction in the risk of death (HR=0.98, 95%CI:0.95-0.999). Can authors identify, why these patients' received antifungals beyond 14 days? was It a persistent ongoing fungal infection or something else? Can authors suggest an optimal duration of antifungal therapy?

 Response: The basic approach to the duration and type of antifungal regimen used was determined according to the attending physician’s judgment. Although an antifungal treatment duration of > 4 weeks was significantly associated with lower mortality than a 2-week course in univariable analysis, the finding was no longer statistically significant after applying multivariable adjustments (S1 Table). This information has been added to the RESULTS AND DISCUSSION section, as below. 

“In sensitivity analyses comparing combination vs. single and 2 weeks vs. > 2-4 (>2, >3, or >4) weeks of antifungal therapy, treatment duration of > 4 weeks was associated with a significantly lower mortality than a 2-week course in univariable analysis. However, the results were no longer statistically significant after applying multivariable adjustments (S1 Table). Combination and single antifungal regimens were comparable with respect to patient survival.”

6) Antifungal choices varied? Only 70% received Amphotericin- B, which was the basis of choosing other antifungals and combinations.

 Response: The prescribed antifungal regimens in our study varied greatly in relation to the choice, duration, and dose of antifungal agent and the use of single or combined therapy, probably due to the lack of specific recommendations made in the 2016 ISPD Peritonitis Guidelines [18]. Moreover, the guidelines do not make specific treatment recommendations according to fungal type. Recommendations that are made are based on studies of Candida peritonitis. Therefore, in the absence of high certainty evidence and clear recommendations, the approach to treatment in this study was somewhat variable. Moreover, antifungal susceptibility tests are limited in Thailand; only 2 (9%) laboratories performed antimicrobial susceptibility tests for fungi [Kanjanabuch et al. Kidney Int Rep 2021]. Our result affirms the existence of treatment variation and demonstrates the association of such variations wtih mortality rate. 

7) The only strength of the article is a large number of fungal episodes, and mortality predictors. Authors need to address the above mentioned comments to improve the manuscript.

 Response: Thank you for your valuable comments.

Journal Requirements:

2) Please provide additional details regarding participant consent. 

3) We note that the grant information you provided in the ‘Funding Information’ and ‘Financial Disclosure’ sections do not match. 

5) Thank you for stating the following in the Acknowledgments/ Disclosure Section of your manuscript. Please note that funding information should not appear in the Acknowledgments section or other areas of your manuscript. We will only publish funding information present in the Funding Statement section of the online submission form. Please remove any funding-related text from the manuscript and let us know how you would like to update your Funding Statement. Please include your amended statements within your cover letter; we will change the online submission form on your behalf.

6) One of the noted authors is a group or consortium [Nephrology Society of Thailand]. In addition to naming the author group, please list the individual authors and affiliations within this group in the acknowledgments section of your manuscript. Please also indicate clearly a lead author for this group along with a contact email address.

Response: The manuscript’s format has been revised accordingly. Please revises the funding statement in the online submission form as shown below.

“This study was supported by the Thailand Science research and Innovation Fund Chulalongkorn University CU_FRB65_hea (19)_026_30_07, Chulalongkorn University, Thailand and the National Research Council of Thailand (156/2560). TK has received consultancy fees from VISTERRA as a country investigator and is a current recipient of the National Research Council of Thailand and received speaking honoraria from Astra Zeneca and Baxter Healthcare. JP has received speaking honoraria from Astra Zeneca, Baxter Healthcare, DaVita Healthcare Partners, Fresenius Medical Care, Dialysis Clinics Incorporated, Satellite Healthcare, and served as a Baxter consultant Healthcare, DaVita Healthcare Partners, Fresenius Medical Care, and LiberDi. DJ has previously received consultancy fees, research grants, speaker's honoraria, travel sponsorships from Baxter Healthcare and Fresenius Medical Care, consultancy fees from AstraZeneca, Bayer, and AWAK, and speaker's honoraria from Ono and BI & Lilly, and travel sponsorships from Amgen. He is also supported by an Australian National Health and Medical Research Council (NHMRC) Leadership Investigator Grant. All other authors have no financial conflicts of interest to declare. All funders and pharmaceutical companies did not have any role in the study design, data collection, data analysis, data reporting, and the decision to submit for publication.” 

We hope that we have satisfactorily addressed the reviewers’ comments. Please do not hesitate to contact us if you require any further clarifications or information. 

Thank you very much for your kind consideration.

Sincerely yours,

Prof. Talerngsak Kanjanabuch, MD

On behalf of the authors

---

## [Decision Letter · Decision Letter 1]

10 May 2022

Predictors and outcomes of peritoneal dialysis-related infections due to filamentous molds (MycoPDICS)

PONE-D-21-30246R1

Dear Dr. Kanjanabuch,

We’re pleased to inform you that your manuscript has been judged scientifically suitable for publication and will be formally accepted for publication once it meets all outstanding technical requirements.

Kind regards,

Rebecca A Hall

Academic Editor

PLOS ONE

Additional Editor Comments (optional):

Reviewers' comments:

Reviewer's Responses to Questions

**Comments to the Author**

1. If the authors have adequately addressed your comments raised in a previous round of review and you feel that this manuscript is now acceptable for publication, you may indicate that here to bypass the “Comments to the Author” section, enter your conflict of interest statement in the “Confidential to Editor” section, and submit your "Accept" recommendation.

Reviewer #1: (No Response)

Reviewer #2: All comments have been addressed

2. Is the manuscript technically sound, and do the data support the conclusions?

Reviewer #1: (No Response)

Reviewer #2: Yes

3. Has the statistical analysis been performed appropriately and rigorously? 

Reviewer #1: (No Response)

Reviewer #2: Yes

4. Have the authors made all data underlying the findings in their manuscript fully available?

Reviewer #1: (No Response)

Reviewer #2: Yes

5. Is the manuscript presented in an intelligible fashion and written in standard English?

Reviewer #1: (No Response)

Reviewer #2: Yes

6. Review Comments to the Author

Reviewer #1: (No Response)

Reviewer #2: (No Response)

7. PLOS authors have the option to publish the peer review history of their article (what does this mean?). If published, this will include your full peer review and any attached files.

Reviewer #1: **Yes: **Dr.Krishnaswamy Sampathkumar

Reviewer #2: **Yes: **Narayan Prasad

---

## [Editor Report · Acceptance letter]

16 May 2022

PONE-D-21-30246R1 

Predictors and outcomes of peritoneal dialysis-related infections due to filamentous molds (MycoPDICS) 

Dear Dr. Kanjanabuch:

I'm pleased to inform you that your manuscript has been deemed suitable for publication in PLOS ONE. Congratulations! Your manuscript is now with our production department. 

Kind regards, 

on behalf of

Dr. Rebecca A Hall 

Academic Editor

PLOS ONE